# Rapid Identification of Main Vegetation Types in the Lingkong Mountain Nature Reserve Based on Multi-Temporal Modified Vegetation Indices

**DOI:** 10.3390/s23020659

**Published:** 2023-01-06

**Authors:** Wenjing Fang, Hongfen Zhu, Shuai Li, Haoxi Ding, Rutian Bi

**Affiliations:** College of Resources and Environment, Shanxi Agricultural University, Taiyuan 030031, China

**Keywords:** nature reserve, vegetation index, optimal feature, random forest

## Abstract

Nature reserves are among the most bio-diverse regions worldwide, and rapid and accurate identification is a requisite for their management. Based on the multi-temporal Sentinel-2 dataset, this study presents three multi-temporal modified vegetation indices (the multi-temporal modified normalized difference *Quercus wutaishanica* index (MTM-NDQI), the multi-temporal modified difference scrub grass index (MTM-DSI), and the multi-temporal modified ratio shaw index (MTM-RSI)) to improve the classification accuracy of the remote sensing of vegetation in the Lingkong Mountain Nature Reserve of China (LMNR). These three indices integrate the advantages of both the typical vegetation indices and the multi-temporal remote sensing data. By using the proposed indices with a uni-temporal modified vegetation index (the uni-temporal modified difference pine-oak mixed forest index (UTM-DMI)) and typical vegetation indices (e.g., the ratio vegetation index (RVI), the difference vegetation index (DVI), and the normalized difference vegetation index (NDVI)), an optimal feature set is obtained that includes the NDVI of December, the NDVI of April, and the UTM-DMI, MTM-NDQI, MTM-DSI, and MTM-RSI. The overall accuracy (OA) of the random forest classification (98.41%) and Kappa coefficient of the optimal feature set (0.98) were higher than those of the time series NDVI (OA = 96.03%, Kappa = 0.95), the time series RVI (OA = 95.56%, Kappa = 0.95), and the time series DVI (OA = 91.27%, Kappa = 0.90). The OAs of the rapid classification and the Kappa coefficient of the knowledge decision tree based on the optimal feature set were 95.56% and 0.95, respectively. Meanwhile, only three of the seven vegetation types were omitted or misclassified slightly. Overall, the proposed vegetation indices have advantages in identifying the vegetation types in protected areas.

## 1. Introduction

The Lingkong Mountain Nature Reserve of China (LMNR) is a national nature reserve, where temperate grassland is interlaced with warm temperate deciduous broad-leaved forestland. A high bio-diversity level at the LMNR is vital for maintaining its ecological stability [1,2]. To build a framework for ecosystem management and planning, the first step is to identify, describe, and classify the vegetation units in the entire landscape system [3]. Thus far, vegetation surveys in the LMNR have been mainly conducted through field sampling on a small scale [4,5,6]. Although these surveys are able to acquire accurate vegetation type information, they are time consuming, labor intensive, and they can hardly cover the entirety of the LMNR, thus, better methods are required to facilitate the rational development, utilization, and protection of regional resources. Advancements in remote sensing technology provide an effective solution to this problem because they enable the rapid acquisition of large-scale data, especially in remote areas [7].

Time series remote sensing imageries have not only the spectral information of a single-phase imagery, but they also have a series of time information [8].The main reason for thus is the seasonal variance of the vegetation’s spectral reflectance, which changes according to the season and the growing stage for each vegetation type [9]. This is of great significance for the extraction of vegetation-type information. Continuous time series remote sensing images with a high temporal resolution can reflect the phenological period of the vegetation [10,11]. Compared with the Landsat-8 and MODIS datasets which are commonly used for remote sensing large-scale vegetation classification, the Sentinel-2 satellite imagery provides a significantly higher spatiotemporal resolution. The A/B double satellites can realize a revisit period of five days. Additionally, the Sentinel-2 dataset has a multispectral band including visible, red-edge (RE), near-infrared (NIR), and short-wave infrared spectra. It is a dataset with three wavebands in the red-edge spectral range, making it suitable for monitoring the vegetation’s health [12,13,14]. Numerous studies have proven that the multi-temporal Sentinel-2 dataset can be used for vegetation-type mapping [15,16,17].

Vegetation indices can be widely used to monitor the land cover changes on various scales by amplifying the spectral differences between the surface features through simple mathematical combinations or the conversion of feature bands [18,19]. From a developmental perspective, the early vegetation indices (e.g., the ratio vegetation index (RVI), the difference vegetation index (DVI), and the normalized difference vegetation index (NDVI)) were underlain by the physical principle that chlorophyll selectively absorbs visible red light for photosynthesis and reflects infrared light. These indices were designed to amplify the differences between the different land cover types [20]. On this basis, the early vegetation indices were improved to minimize the influence of the atmospheric and soil background by changing the combination of the red band and NIR band or by introducing the blue band [21,22,23,24,25] (e.g., the soil adjusted vegetation index (SAVI), the enhanced vegetation index (EVI), and the atmospherically resistant vegetation index (ARVI)). However, most vegetation types share a similar reflectance mode, wherein the visible red light is absorbed, and the infrared light is reflected, thus, the classification methods based on the above vegetation indices cannot provide accurate results. To solve this problem, in recent years, researchers have developed a few vegetation indices for identifying certain vegetation types according to the special physiological and growth characteristics of the identification objects. For example, the mangrove vegetation index (MVI) proposed by Baloloy et al. [26] can distinguish mangroves from other types of terrestrial vegetation according to the greenness and humidity of the land cover monitored by the Sentinel-2 green, near-red, and short-wave infrared bands. Meanwhile, d’Andrimont et al. [27] found that rape flowers turn yellow in the booming period; their color features are most obvious in the most vigorous blooming period, and their color features are diminished in the withering process. Based on the NDVI and blue and green bands, they proposed the normalized difference yellow index (NDYI) to extract spatial distribution information about oilseed rape by accurately detecting the peak blooming period. The above vegetation indices can improve the accuracy of remote sensing vegetation classification by modifying the vegetation indices in two ways (i.e., by changing the feature bands and the combinations), however, they have not considered using phenological information. When vegetation indices are used for remote sensing vegetation identification or classification, the bands of the remote sensing images used for developing vegetation indices are still derived from the same time phase. Moreover, the identification targets are undiversified, and the surface cover of the study area is relatively simple in the above studies. Owing to the fragmented land and diverse vegetation types, the identification of vegetation types in the LMNR is much more complex. This study aims to develop a set of modified vegetation indices, some of which are multi-temporal modified vegetation indices containing phenological information. These modified vegetation indices can be used to quickly and accurately investigate the vegetation status in the LMNR, while providing a technical means for ensuring the effective management and scientific utilization of regional vegetation resources.

## 2. Materials and Methods

### 2.1. Profile of the Study Area

The LMNR stretches from 36°33′28″ N to 36°42′52″ N latitudinally and from 111°59′27″ E to 112°07′48″ E longitudinally (Figure 1), with a total operation area of 10,116.8 km^2^. The LMNR is dominated by a warm temperate continental monsoon climate, with an average annual temperature of 6.2 °C, an average daily temperature of at least 10 °C, an annual cumulative temperature of 3000 °C, an annual sunshine duration of 2600 h, an annual plant growing period of between 110 and 125 days, an annual average precipitation of 662 mm (concentrated in July, August, and September, collectively accounting for 74.8% of it), and a frost-free period of approximately 145 days. Based on the results of repeated field survey and relevant forestry reference data, the main vegetation types in the LMNR can be classified into crops, scrub grass, *Pinus tabuliformis*, shaw, *Larix principis-rupprechtii*, pine-oak mixed forests, and *Quercus wutaishanica*.

### 2.2. Data sources

#### 2.2.1. Multi-Temporal Remote Sensing Data

We selected six Sentinel-2 L2A images (dated from 19 February, 25 April, 4 June, 28 August, 17 October, and 21 December 2021) covering the study area, with less than 5% cloudage. The Sentinel-2 remote sensing data were provided by the data center (https://scihub.copernicus.eu/dhus/, accessed on 28 April 2021) of the European Space Agency (ESA), and the L2A-class product data were radiometrically calibrated and atmospherically corrected. In this study, Bands 1, 9, and 10 were excluded due to their low spatial resolution (60 m), and all of the images with a 20 m resolution were re-sampled into images with a 10 m resolution through nearest neighbor interpolation.

#### 2.2.2. Field Survey Data

To better investigate the spatial distribution of the vegetation communities, the researchers conducted a field survey of the primary vegetation types in the LMNR from 10 September to 15 September 2021 (Figure 2). Additionally, some areas that were inaccessible to humans were aerially photographed using unmanned aerial vehicles.

Given that mountainous topographical factors (e.g., slope, slope direction, and elevation) may affect the radiation signals recorded by satellite sensors [28,29], the sampling points were selected at different altitudinal gradients, slopes, and slope directions through the three-dimension terrain by combining the remote sensing images and DEM data. Each sample area comprised 3 × 3 pixels (30 × 30 m in size), and each sampling point was mainly selected for a single vegetation type. Finally, 270 samples under seven major vegetation types were drawn, involving a total of 2430 pixels. The field survey data were used for training and verification in the ratio of 6:4.

### 2.3. Methodology

#### 2.3.1. Classification Features

##### Typical Vegetation Indices

Based on the study purpose and band information in the Sentinel-2 dataset, three typical vegetation indices (including time series RVI, DVI, and NDVI) were selected to evaluate their ability to identify the vegetation types in the LMNR.

Proposed by Jordan in 1969, the RVI is generally used to assess the biomass, and it is extremely sensitive to vegetation when the vegetation cover is high [30]. Its calculation equation is as follows:RVI = RED/NIR.(1)

Meanwhile, the DVI is simple in its structure [31]. It is sensitive to changes in the soil background, and its calculation equation is as follows:DVI = NIR − RED.(2)

The NDVI is the most commonly used vegetation index [32]. The NDVI is calculated through a normalization process, so it is sensitive to green vegetation even in areas with low vegetation coverage. Its calculation equation is as follows:NDVI = (NIR − RED)/(NIR + RED).(3)

NIR and RED, respectively, denote the reflectance values of the NIR and red bands.

##### Uni/Multi-Temporal Modified Vegetation Indices

Based on the acquired Sentinel-2A image data for six periods, the average spectral values of seven vegetation types were statistically analyzed (Figure 3). The spectral differences between the surface features were analyzed according to the spectral feature curves. During February and December, the spectral curve of *Pinus tabuliformis*, which is an evergreen plant, exhibited unique vegetation characteristics such as “two valleys and one peak” and “red edge” [19]. In the Sentinel-2 band, this was represented by the two absorption valleys such as “red valley” and “blue valley”, which were formed by the vegetation near the red and blue bands. Near the green band, a reflection peak was formed, owing to the strong reflection effect on green light, which is also referred to as “green peak”. This feature is due to the influence of chlorophyll, which has a strong absorption effect on blue and red lights and a strong reflection effect on green light. The spectral values of vegetation in the RE2, RE3, and NIR bands sharply increased, forming a nearly straight curve with a large slope, which is commonly known as “red-edge”. This feature depends on the cell structure of the leaves. The other vegetation types are in a dormancy state, with their leaves being withered or with their vegetation spectral features being diminished and their spectral curves tending to be flat, with no evident “peaks” and “valleys”; otherwise, the spectral features are similar to those of bare soil, and they are represented in the Sentinel-2 band as a reflection peak near the short-wave infrared (SWIR). Therefore, there exist significant spectral differences between *Pinus tabuliformis* and the other vegetation types. Similarly, as the sowing of major crops did not start in June, there was no vegetation growth in this area. Therefore, the spectral features of its area should be similar to those of bare soil. In contrast, other vegetation had already entered the vigorous growth period, and their spectral curves exhibited evident features of “two valleys and one peak” and “red edge”. Therefore, there exist significant spectral differences between the major crops and other vegetation types. These spectral differences occur because the spectrum of the vegetation growth period is unique and significant, which can be represented by the Sentinel-2 red band and NIR band. These typical vegetation indices were developed for the advantage of increasing the wavelength from visible red light to reflected infrared, which is caused by the selective absorption of red light by chlorophyll for photosynthesis in the vegetation spectrum. Therefore, *Pinus tabuliformis* and crops can be easily identified using these typical vegetation indices. In the six time-phases that were selected in this study, scrub grass, *Quercus wutaishanica*, pine-oak mixed forests, *Larix principis-rupprechtii*, and shaw had relatively similar reflectance values in the red and NIR bands. Therefore, high-precision remote sensing monitoring and identification with typical vegetation indices are difficult. Thus, to improve the classification accuracy of remote sensing vegetation in the LMNR, it is of vital importance to develop a set of modified vegetation indices that can effectively amplify the spectral differences among the scrub grass, *Quercus wutaishanica*, pine-oak mixed forests, *Larix principis-rupprechtii*, and shaw.

Expanding the inter-class spectral differences and reducing the intra-class differences were the basic principles in developing our modified vegetation indices. Considering the spectral similarities in the five vegetation types that were to be identified, identifying them using a certain vegetation index was difficult, however, the hierarchical classification can effectively improve the classification accuracy [33]. Therefore, in a sequence from difficult to easy, we developed the indices one by one for specific vegetation types and identified them sequentially. First, through a spectral analysis, we selected the vegetation type with unique spectral features, which was the one with the highest spectral difference among the other vegetation types that were to be identified, which was the object of developing the modified vegetation index. This spectral difference can be represented by dual-band combinations of the same temporal phase or those of different temporal phases, and these dual-band combinations are referred to as feature band combinations. The vegetation may have multiple sets of feature band combinations. Based on standard deviations, multiple sets of feature band combinations were screened, and the band combination with the most stable inter-class difference (i.e., the band combination with the smallest sum of standard deviations) was selected. The calculation formula of the modified vegetation index was determined based on the relationship between the spectral values of the feature dual bands and on the band combinations of typical vegetation (Figure 4).

In this study, we developed four uni (multi)-temporal modified vegetation indices for pine-oak mixed forests, *Quercus wutaishanica*, scrub grass, and shaw, and the development process of the four indices is described below.

Regarding, the uni-temporal modified difference pine-oak mixed forest index (UTM-DMI), in February and December, the spectral values of the RE4 band of pine-oak mixed forests are basically the same as those of the SWIR-1 band, and the spectral values of the RE4 band of other vegetation types that were to be identified are significantly lower than those of the SWIR-1 band. This is represented by the reflection peaks near the SWIR-1 and SWIR-2 bands. Therefore, the RE4 and SWIR-1 bands in February and the RE4 and SWIR-1 bands in December are combinations of feature bands that are used for identifying the pine-oak mixed forest images. Based on standard deviations, the two band combinations were further screened, and the band combination with the most stable intra-class variance (i.e., the band combination with the smallest sum of standard deviations) was selected, namely, the RE4 and SWIR-1 bands in February (Table 1). Based on the principle for constructing the DVI, the UTM-DMI was constructed by calculating the spectral difference between the RE4 band and the SWIR-1 band in February.
UTM-DMI = RE4_&Feb_ − SWIR-1_&Feb_,(4)
where RE4_&Feb_ denotes the reflectance of the RE4 band in February, and SWIR-1_&Feb_ denotes the reflectance of the SWIR-1 in February.

Here, the vegetation types that were to be identified include *Quercus wutaishanica*, scrub grass, *Larix principis-rupprechtii*, and shaw.

Regarding the multi-temporal modified normalized difference *Quercus wutaishanica* Index (MTM-NDQI), in June and October, the spectral values of the NIR and RE4 bands of *Quercus wutaishanica* were significantly higher than those of other vegetation types that were to be identified. This is due to the internal structure and density of the *Quercus wutaishanica* leaves. Therefore, the NIR band in June and October and the RE4 band in June and October are the combinations of feature bands that were used for identifying the *Quercus wutaishanica* images. Based on standard deviations, the two band combinations were further screened, and the band combination with the most stable intra-class variance was selected, namely, the RE4 band in June and October. For different vegetation types, the spectral value of the RE4 band is smaller than 0.54 in June, and it is smaller than 0.34 in October, with their sum being smaller than 1 (Table 2). The vegetation index variability can be amplified by increasing the numerator value and reducing the denominator value. Based on the principle for constructing the NDVI, the MTM-NDQI was determined using the following equation:MTM-NDQI = (1 − RE4_&June_ − RE4_&Oct_)/(RE4_&June_ + RE4_&Oct_),(5)
where RE4_&June_ denotes the reflectance of the RE4 band in June, and RE4_&Oct_ denotes the reflectance of the RE4 band in October.

Here, the vegetation types that were to be identified include scrub grass, *Larix principis-rupprechtii*, and shaw.

Regarding the multi-temporal modified difference scrub grass index (MTM-DSI), from June to August, the spectral values of the RE2, RE3, NIR, and RE4 bands of the scrub grass increased, whereas those of other vegetation types that were to be identified decreased. Therefore, the RE2, RE3, NIR, and RE4 bands in June and August constitute four combinations of feature bands for identifying scrub grass images. Based on standard deviations, the four band combinations were further screened, and the band combination with the most stable intra-class variance was selected, namely, the RE2 band in June and August (Table 3). Based on the principle for constructing the DVI, the MTM-DSI was constructed by calculating the spectral difference between the RE2 band in June and August:MTM-DSI = RE2_&June_ − RE2_&Aug_,(6)
where RE2_&June_ denotes the reflectance of the RE2 band in June, and RE2_&Aug_ denotes the reflectance of the RE2 band in August.

Here, the vegetation types that were to be identified include *Larix principis-rupprechtii* and shaw.

Regarding the multi-temporal modified ratio shaw index (MTM-RSI), the spectral values of the RE2, RE3, NIR, and RE4 bands of shaw are larger than those of *Larix principis-rupprechtii* in June, but they are smaller than those of *Larix principis-rupprechtii* in October, indicating that from June to October, the spectral values of these bands of shaw changed more significantly than those of *Larix principis-rupprechtii*. Therefore, the RE2, RE3, NIR, and RE4 bands in June and October constitute four combinations of feature bands for identifying shaw images. Based on standard deviations, the four band combinations were further screened, and the band combination with the most stable intra-class variance was selected, namely, the RE2 band in June and October (Table 4). The vegetation index variability can be amplified by increasing the numerator value and reducing the denominator value. Based on the principle for constructing the RVI, the MTM-RSI was constructed by dividing the spectral difference between the RE2 band in June and October and the spectral value of the RE2 band in October:MTM-RSI = (RE2_&June_ − RE2_&Oct_)/RE2_&Oct_,(7)
where RE2_&June_ denotes the reflectance of the RE2 band in June, and RE2_&Oct_ denotes the reflectance of the RE2 band in October.

##### Optimal Feature Set

Figure 5 shows a box plot of typical vegetation indices for seven sample types in the six periods. The overall separability of the NDVI samples is the most significant in February, April, June, and December, and the overall separability of DVI samples is the most significant in August and October. In February, April, and June, the NDVI of crops is lower than that of other vegetation types, while in April, the NDVI of crops tends to overlap less often with that of other vegetation types, making it suitable to identify crop images. In February and December, the NDVI of *Pinus tabuliformis* is significantly higher than that of other vegetation types, while in December, the NDVI outliers of *Pinus tabuliformis* are less common, making it suitable to identify *Pinus tabuliformis* images. In each time phase, the typical vegetation indices of any of the five other vegetation types (including scrub grass, *Quercus wutaishanica*, pine-oak mixed forests, *Larix principis-rupprechtii*, and shaw) overlap with another vegetation type; the overlapping samples account for more than 25% of the total samples, with low separability being observed between them.

In summary, the NDVI in December and the NDVI in April can easily identify *Pinus tabuliformis* and crops. Hence, the NDVI in December, the NDVI in April, and the UTM-DMI, MTM-NDQI, MTM-DSI, and MTM-RSI constitute the optimal feature set for remote sensing vegetation classification in the LMNR.

#### 2.3.2. Classification Method and Accuracy Evaluation

The random forest algorithm is a machine learning algorithm that integrates multiple regression decision trees based on the principle of ensemble learning [34]. It is characterized by high accuracy and high stability, while being independent of the high-dimension space, thus, it is widely used for classification or feature selection. Its performance depends on two major parameters, including the number of decision trees and the number of features involved in each decision tree [35], which were, respectively, set to 250 and the square root of the total number of inputted feature variables.

Unlike other types of decision trees or machine learning algorithms, knowledge decision trees can generate classification rules only through the summarization of expert experience, inductive method, and mathematical statistics rather than lots of sample training. Knowledge decision trees offer a convenient and effective solution to land cover classification [36,37]. Based on a statistical analysis of the optimal features set for the seven vegetation types, appropriate thresholds were selected experientially to construct a knowledge decision tree model for vegetation classification in the LMNR (Figure 6).

In this study, the pixel-based confusion matrix method was used to evaluate the accuracy of the classification results. As the most commonly used evaluation method in image classification studies, this method adopted a standard accuracy evaluation matrix with n rows and n columns. The overall accuracy (OA) and Kappa coefficient are evaluation indices for the overall classification effect, and the user accuracy (UA) and producer accuracy (PA) are evaluation indices for the classification effect of different vegetation types [38].
(8)$$\mathrm{OA}=\frac{\sum_{i=1}^{k}N_{ii}}{N}$$
(9)$$\mathrm{PA}=\frac{N_{kk}}{N_{+k}}$$
(10)$$\mathrm{UA}=\frac{N_{kk}}{N_{k+}}$$
(11)$$\mathrm{Kappa}=\frac{N\sum_{i=1}^{k}N_{ii}-\sum_{i=1}^{k}\left(N_{i+}\times N_{+i}\right)}{N^{2}-\sum_{i=1}^{k}\left(N_{i+}\times N_{+i}\right)}$$

Here, *k* denotes the total number of vegetation types; N denotes the total number of samples; the diagonal element N_*kk*_ denotes the number of correctly classified real samples; N_*k*+_ denotes the total number of samples under Type *i*; N_+*k*_ denotes the number of samples classified as Type *j*; N*_i_* and N*_j_* denote the value associated with the element (*i*, *j*) in the confusion matrix, respectively.

## 3. Results

### 3.1. Random Forest Classification by Different Classification Feature Sets

In this study, the random forest algorithm was used to classify the optimal feature set, time series RVI, DVI, and NDVI. As described in Table 5, the optimal feature set has the highest OA and Kappa coefficient (98.41% and 0.98, respectively), which is followed by the time series NDVI (96.03% and 0.95), the time series RVI (95.56% and 0.95), and the time series DVI (91.27% and 0.90). In other words, the OA and Kappa coefficient of the optimal feature set are higher than those of the time series NDVI, RVI, and DVI, respectively, by 2.38%, 2.86% and 7.14%, as well as 0.03, 0.03 and 0.08. For the classification effect of different vegetation types, only *Pinus tabuliformis* obtains the highest classification accuracy in terms of the time series DVI, with the PA and UA being 100% and 100%, respectively. Crops, scrub grass, *Quercus wutaishanica*, pine-oak mixed forests, *Larix principis-rupprechtii*, and shaw all obtained the highest classification accuracy in terms of the optimal feature set, with their PA and UA values being 100%, 98.61%, 99.21%, 97.44%, 95.83%, and 96.83%, as well as 98.44%, 100%, 97.66%, 100%, 100%, and 93.85%, respectively. Evidently, the overall classification ability of the optimal feature set is higher than that of the time series RVI, DRI, and NDVI.

### 3.2. Rapid Classification by the Decision Tree Based on the Optimal Feature Set

In this study, *Pinus tabuliformis*, crops, pine-oak mixed forests, *Quercus wutaishanica*, scrub grass, and *Larix principis-rupprechtii* were hierarchically identified using the above knowledge decision tree model. Figure 7 shows the classification of vegetation types in the LMNR. As dominant species in the LMNR, *Pinus tabuliformis*, *Quercus wutaishanica,* and pine-oak mixed forests cover the largest area of the LMNR, and they are distributed in all directions of the LMNR. These three vegetation types are followed by crops and scrub grass; scrub grass is usually located around crops, and it is distributed in valley areas of the northeastern and central LMNR. *Larix principis-rupprechtii* and shaw cover a small area, and they are sporadically distributed in the pine-oak mixed forests.

Table 6 lists the confusion matrix results for vegetation classification. Within the 630 selected verification points, there are 62 real crop samples, which have all been correctly classified. There are 72 real scrub grass samples, which have all been correctly classified. There are 117 real *Pinus tabuliformis* samples, which have all been correctly classified. There are 135 real *Quercus wutaishanica* samples, among which 125 samples have been correctly sampled and 10 samples have been misclassified. There are 125 real pine-oak mixed forest samples, among which 108 samples have been correctly sampled and 17 samples have been misclassified. There are fifty-nine real *Larix principis-rupprechtii* samples, among which fifty-eight samples have been correctly sampled and one sample has been misclassified. There are 60 real shaw samples, which have all been correctly classified.

In terms of the UA and PA, the classification accuracy of all of the vegetation types is higher than 80%. The classification accuracy of scrub grass and *Pinus tabuliformis* is the highest, which are followed by crops, shaw, *Quercus wutaishanica*, and *Larix principis-rupprechtii*, whereas the classification accuracy of pine-oak mixed forests is the lowest. This is due to the fact that pine-oak mixed forests comprise associated deciduous trees (e.g., Pterocarya stenoptera and Betula) in addition to the dominant *Pinus tabuliformis* and *Quercus wutaishanica*. The differences in the associated tree species or the mixing ratio will increase their inter-class variation, and they will overlap with other vegetation types, thus resulting in misclassification and omission. The OA and Kappa coefficient are, respectively, 95.56% and 0.95, respectively, indicating a good classification effect.

## 4. Discussion

Remote sensing vegetation classification based on vegetation indices is essentially conducted to amplify the spectral difference between the target vegetation and other surface features and highlight the spectral features of the target vegetation by performing spectral changes or the enhancement of the spectral information contained in the bands. In this study, feature bands were selected based on the special physiological and growth patterns of specific identification objects. This is the underlying principle which explains why multi-temporal modified vegetation indices can improve the accuracy of vegetation classification. The three multi-temporal modified vegetation indices were developed based on the RE2 and RE4 bands within the red-edge range [39,40,41]. Many studies have shown that the “red-edge” information of vegetation can effectively reflect the growth and health status of vegetation. For both of the uni-temporal modified vegetation indices developed in this study and the typical vegetation indices developed in previous studies, dimensionality reduction is conducted in the dimension of spectral space through a combination of uni-temporal bands. For multi-temporal modified vegetation indices, dimension reduction is conducted on multi-dimensional spectral features in two dimensions (i.e., the temporal and spectral space dimensions). Because of the changes in the plant morphology and chlorophyll concentration in different growth periods (e.g., sprouting, leaf growth, and maturity), vegetation exhibits different reflectance features in remote sensing images of different phases. Through the combination of multi-temporal bands of the target vegetation, phenological information can be introduced and projected onto a clearly identifiable low-dimension value range without the need for additional data, further improving the accuracy of classification.

It is worth mentioning that the classification accuracy of the decision tree algorithm is lower than that of the random forest algorithm in this study when the optimal feature set was used as the classification features. However, this does not mean that the random forest algorithm is superior to the decision tree algorithm. Most studies argued that other machine learning algorithms (e.g., support vector machines, random forest, and neural network ones) provide a high classification accuracy [42,43,44], but they are time consuming, and their success depends partly on large amounts of training and quality of generated training data, thus requiring careful sampling schemes. This situation is more severe in complex or mountainous landscapes with limited accessibility or uneven distribution of land cover types [45]. During the development of decision tree rule sets, thresholds can be established in diverse ways (e.g., expert knowledge, trial and error, and binary recursive decision trees) [46]. The threshold values of vegetation indices are definite and directly available, so the knowledge decision tree classification based on vegetation indices can be achieved simply and rapidly. This explains why many studies prefer to use spectral indices combined with the knowledge decision tree algorithm [47,48,49].

The “salt and pepper phenomenon” in the classification results is not only associated with surface feature fragmentation in the study area, but it is also influenced by pixel-based classification. In future studies, decision tree classification will be used in conjunction with object-oriented classification, and multi-temporal spectral information will be used in conjunction with texture information for developing vegetation indices [50].

## 5. Conclusions

Based on the multi-temporal Sentinel-2 dataset, we developed three multi-temporal modified vegetation indices to improve the classification accuracy of the remote sensing of vegetation in the LMNR. Based on the advantages of typical vegetation indices (e.g., amplifying spectral differences and dimensionality reduction of the spectral space), the multi-temporal modified vegetation indices integrate the phenological information of regional vegetation, thus performing better in remote sensing vegetation classification. Compared with machine learning algorithms (e.g., random forest), knowledge decision tree classification based on vegetation indices simplifies the classification process and improves its operability, making it a suitable method for regional-scale vegetation monitoring. Based on the advantages of vegetation indices and multi-temporal remote sensing data, this study developed multi-temporal modified vegetation indices, thus improving the conventional method for developing vegetation indices based only on uni-temporal bands. The findings of this study provide a new approach to developing vegetation indices that can improve the accuracy of remote sensing vegetation classification based on multi-temporal remote sensing data.

## Figures and Tables

**Figure 1 sensors-23-00659-f001:**
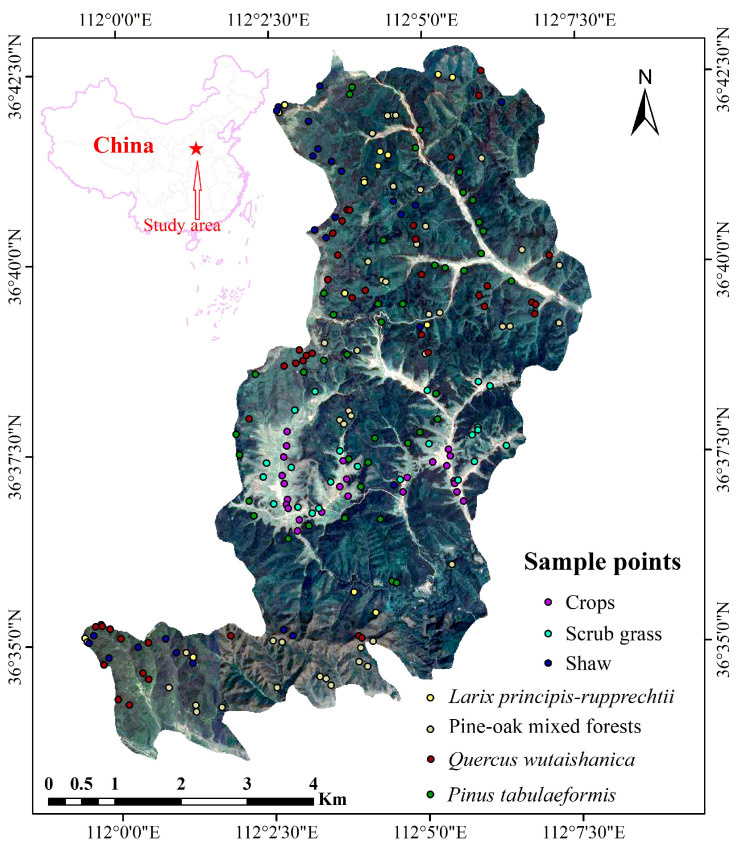
Location of the study area.

**Figure 2 sensors-23-00659-f002:**
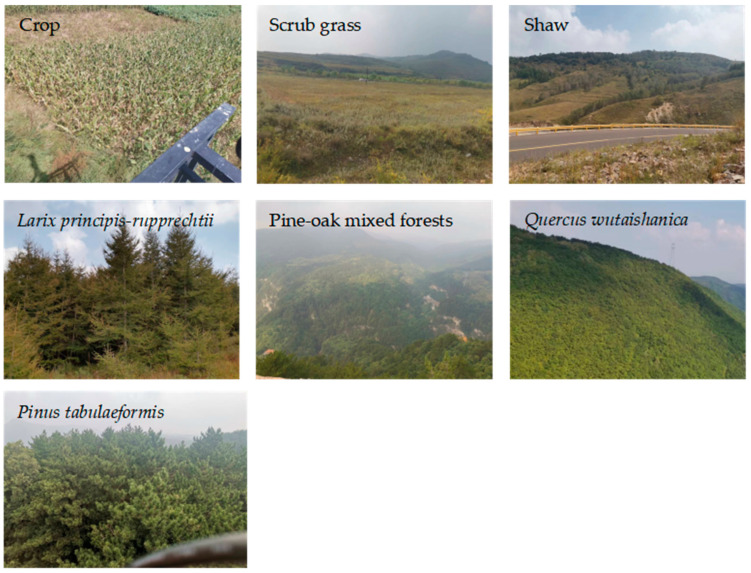
Seven main vegetation types in the LMNR.

**Figure 3 sensors-23-00659-f003:**
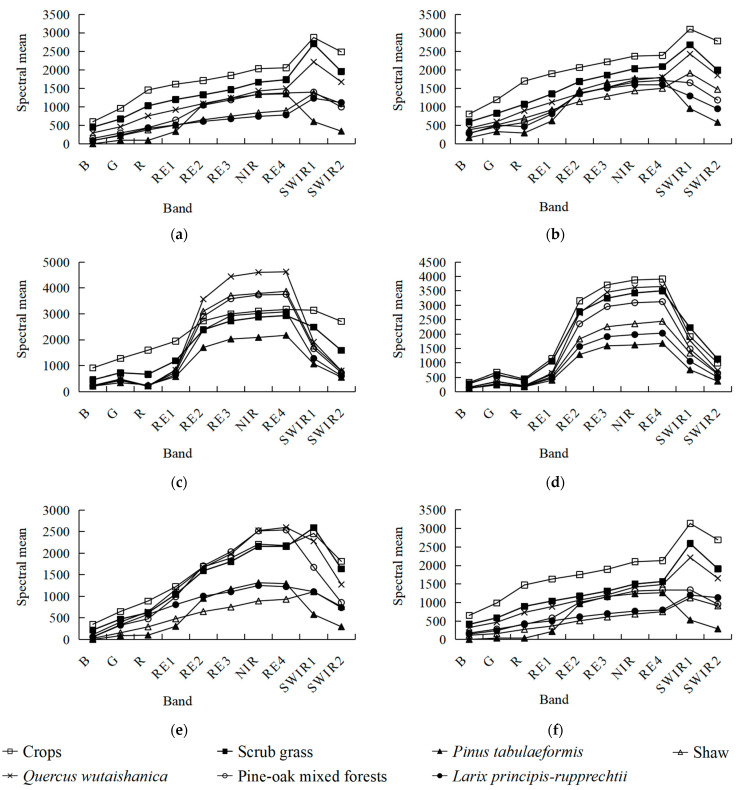
Spectral feature curve of vegetation types on (**a**) 19 February, (**b**) 25 April, (**c**) 4 June, (**d**) 28 August, (**e**) 17 October, (**f**) and 21 December.

**Figure 4 sensors-23-00659-f004:**
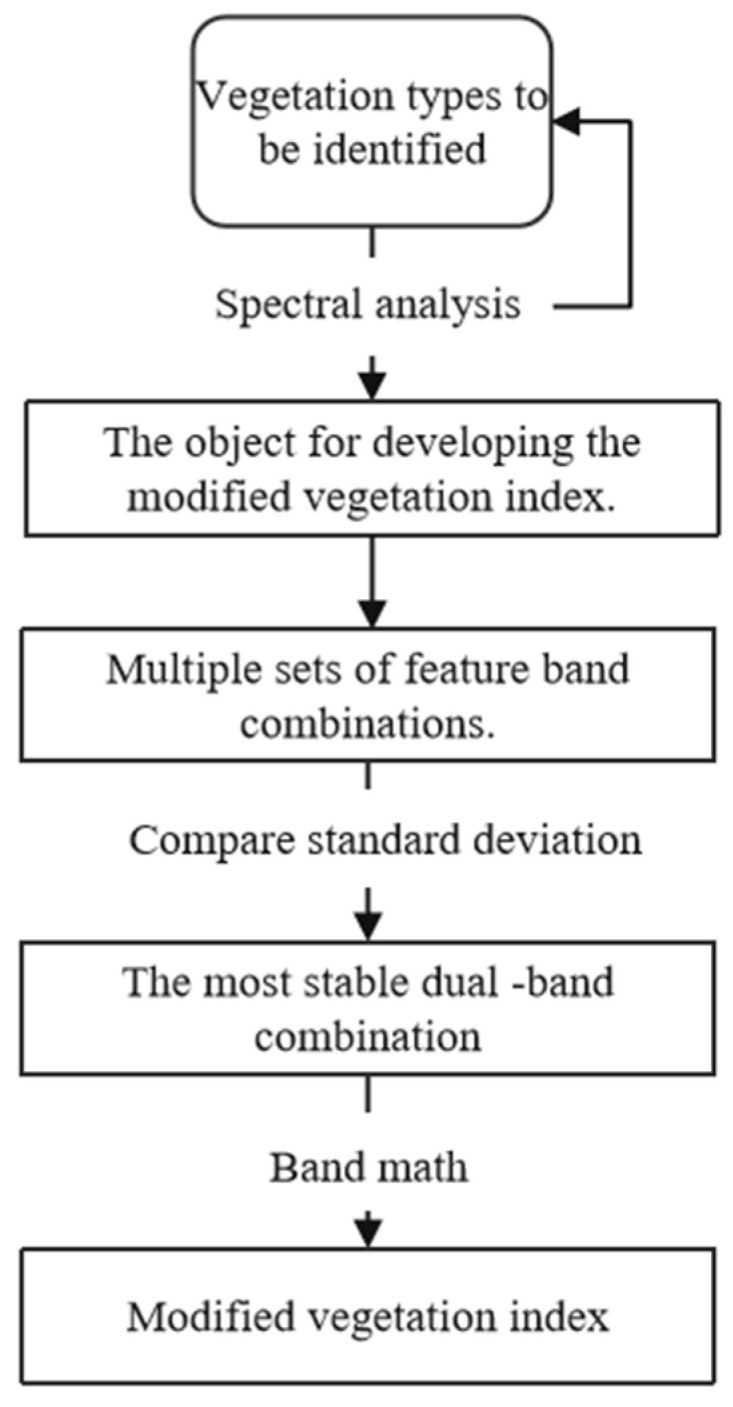
Flow chart for developing the modified vegetation index.

**Figure 5 sensors-23-00659-f005:**
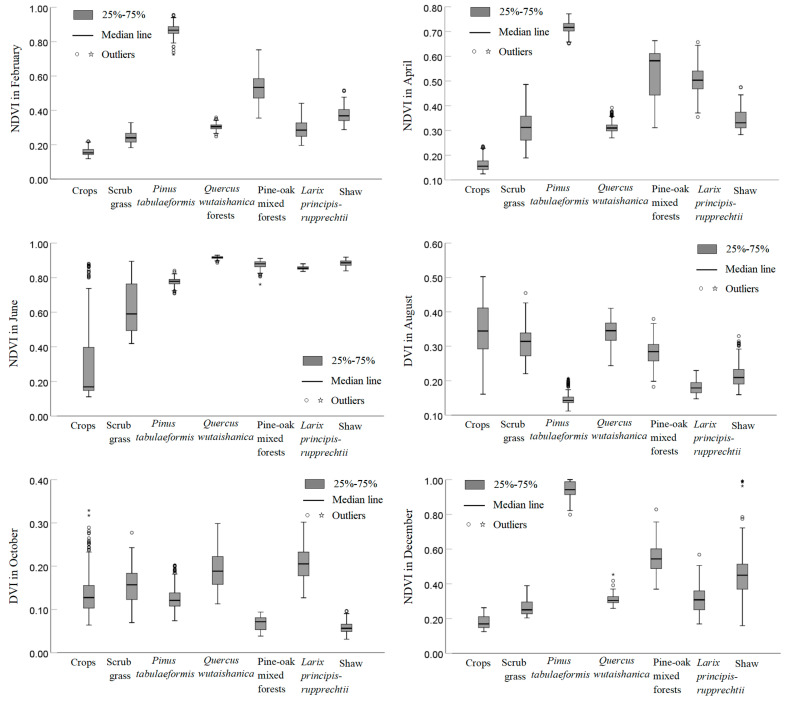
Box plot of typical vegetation indices.

**Figure 6 sensors-23-00659-f006:**
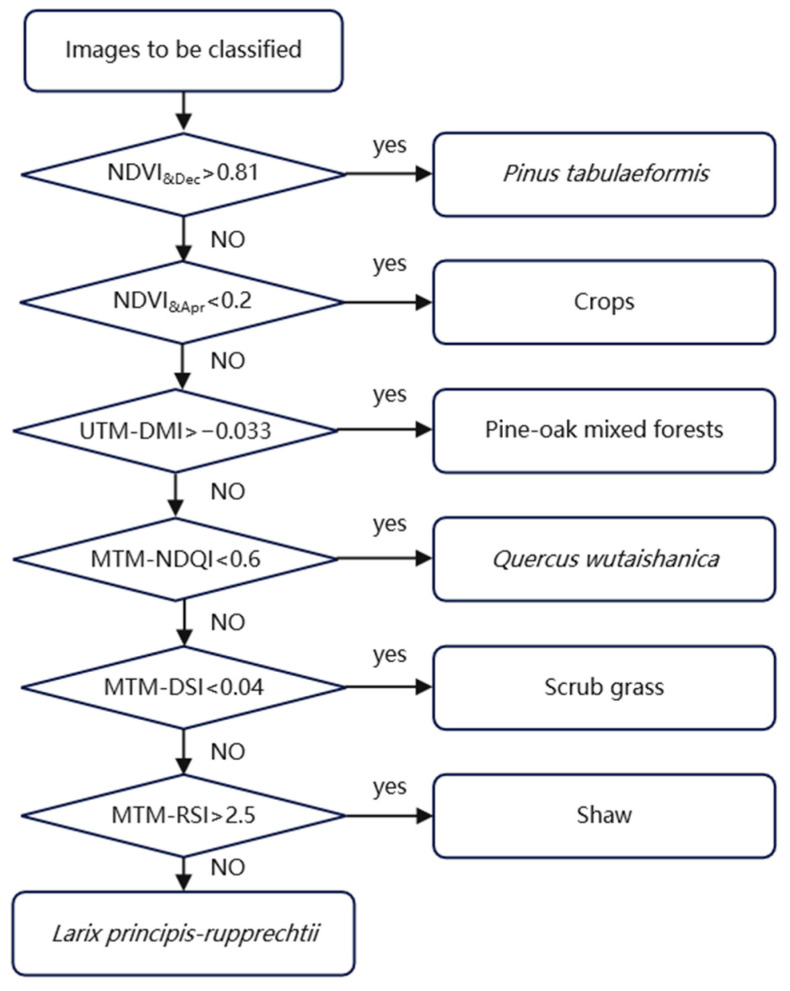
Model for knowledge decision tree.

**Figure 7 sensors-23-00659-f007:**
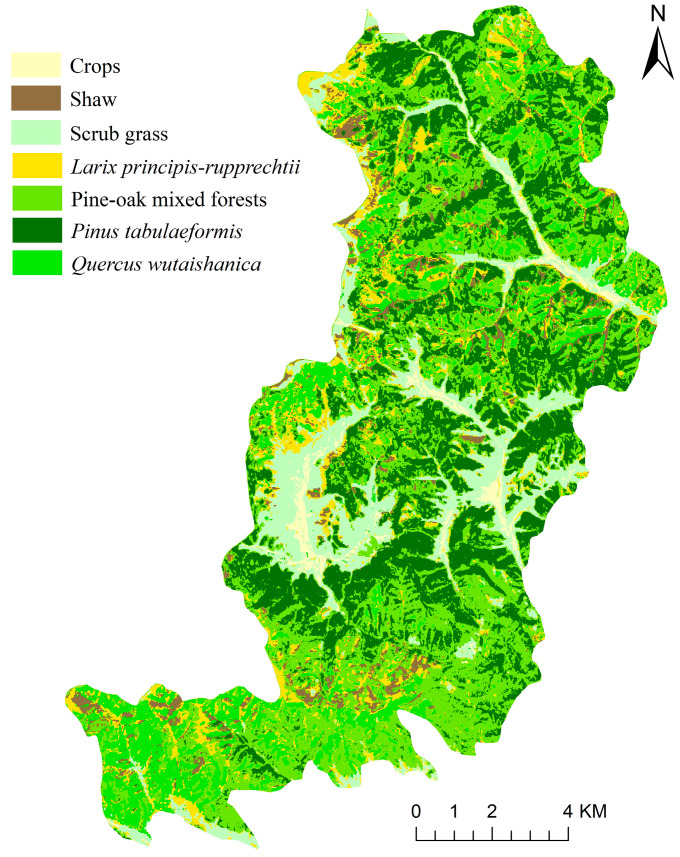
Classification of main vegetation types in the LMNR.

**Table 1 sensors-23-00659-t001:** Sum of standard deviations of feature band combinations for pine-oak mixed forests.

Feature Band Combinations	Sum of Standard Deviations
RE4_&Feb_ and SWIR-1_&Feb_	391.40
RE4_&Dec_ and SWIR-1_&Dec_	456.15

**Table 2 sensors-23-00659-t002:** Sum of standard deviations of feature band combinations for *Quercus wutaishanica*.

Feature Band Combinations	Sum of Standard Deviations
NIR_&June_ and NIR_&Otc_	720.60
RE4_&June_ and RE4_&Otc_	719.15

**Table 3 sensors-23-00659-t003:** Sum of standard deviations of feature band combinations for scrub grass.

Feature Band Combinations	Sum of Standard Deviations
RE2_&June_ and RE2_&Aug_	724.87
RE3_&June_ and RE3_&Aug_	897.84
NIR_&June_ and NIR_&Aug_	976.94
RE4_&June_ and RE4_&Aug_	877.05

**Table 4 sensors-23-00659-t004:** Sum of standard deviations of feature band combinations for shaw.

Feature Band Combinations	Sum of Standard Deviations
RE2_&June_ and RE2_&Oct_	545.04
RE3_&June_ and RE3_&Oct_	683.33
NIR_&June_ and NIR_&Oct_	785.02
RE4_&June_ and RE4_&Oct_	743.65

**Table 5 sensors-23-00659-t005:** Classification accuracy of feature set for different vegetation indices.

Vegetation Type	The Optimal Feature Set		Multi-Temporal DVI		Multi-Temporal RVI		Multi-Temporal NDVI	
	PA/%	UA/%	PA/%	UA/%	PA/%	UA/%	PA/%	UA/%
Crops	100	98.44	100	81.82	100	90	100	94.03
Scrub grass	98.61	100	90.28	98.48	90.28	100	94.44	98.55
*Pinus tabulaeformis*	100	98.32	100	100	100	98.32	100	100
*Qrcus wutaishanica*	99.21	97.66	93.65	88.72	99.21	93.28	98.41	94.66
Pine-oak mixed forests	97.44	100	87.18	100	97.44	99.13	94.87	100
*Larix principis-rupprechtii*	95.83	100	69.44	94.34	90.28	100	91.67	100
Shaw	96.83	93.85	95.24	73.17	84.13	85.48	88.89	81.16
OA/%	98.41		91.27		95.56		96.03	
kappa	0.98		0.90		0.95		0.95	

**Table 6 sensors-23-00659-t006:** Confusion matrix for the classification results of the knowledge decision tree.

Class	Ground Truth (pixel)								PA
	Veg 1	Veg 2	Veg 3	Veg 4	Veg 5	Veg 6	Veg 7	Total	%
Veg 1	62	0	0	0	0	0	0	62	98.41
Veg 2	0	72	0	0	0	0	0	72	100
Veg 3	0	0	117	0	0	0	0	117	100
Veg 4	1	0	0	125	9	0	0	135	99.21
Veg 5	0	0	0	0	108	14	3	125	92.31
Veg 6	0	0	0	1	0	58	0	59	80.57
Veg 7	0	0	0	0	0	0	60	60	95.24
Total	63	72	117	126	117	72	63	630	-
UA%	100	100	100	92.59	86.4	98.31	100	-	-
OA: 95.56% kappa: 0.95									

Veg 1. Crops; Veg 2. Scrub grass; Veg 3. *Pinus tabulaeformis*; Veg 4. *Quercus wutaishanica*; Veg 5. Pine-oak mixed forests; Veg 6. *Larix principis-rupprechtii*; Veg 7. Shaw.

## Data Availability

Not applicable.

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
