# Peer review of "Rapid Identification of Main Vegetation Types in the Lingkong Mountain Nature Reserve Based on Multi-Temporal Modified Vegetation Indices"

_sensors, 2023, doi:10.3390/s23020659_

Round 1

Reviewer 1 Report

This study provides the rapid identification of main vegetation types based on multi-temporal modified vegetation indices. The significance of the study’s content is good, and the quality of presentation is good. The following is the specific suggestions:

1. Line 158-159, what are “significant spectral differences”? Please provide the specific differences.

2. The same problems exist in Line 161-162. What are “ obvious spectral differences”? Please provide the specific differences.

3. Section 2.3.1.2, the classification method is not clear. Please provide the overall classification method.

4. The text size is not uniform in Figure 2.

5. The legends in Figure 3 are confused, and not distinguishable.

6. Line 194, Table 4 may be Table 1.

Author Response

Thank you for your valuable comments! I have responsed to them point-by-point in the attachment.

Reviewer 2 Report

The manuscript titled “Rapid Identification of Main Vegetation Types in the Lingkong Mountain Nature Reserve Based on Multi-temporal Modified Vegetation Indices” presents three multi-temporal modified vegetation indices based on the multi-temporal Sentinel-2 dataset to improve the classification accuracy of remote-sensing vegetation in the Lingkong Mountain Nature Reserve of China. From my viewpoint, the methods used are appropriate and well chosen for the topic. The work done is well structured and well written with interesting findings. The topic fits the scope of this journal. However, I see some points that need improvement in the manuscript, before being published. Therefore, I suggest some major revisions be done.

The comments in the PDF justify my recommendation to accept this paper after a 'minor' revision. I hope this encourages the authors to improve this interesting study.

Author Response

(The authors gave the same response as above.)
